# Anti-Inflammatory Drug Candidates for Prevention and Treatment of Cardiovascular Diseases

**DOI:** 10.3390/ph16010078

**Published:** 2023-01-04

**Authors:** Quentin Delbaere, Nicolas Chapet, Fabien Huet, Clément Delmas, Nathan Mewton, Fabrice Prunier, Denis Angoulvant, François Roubille

**Affiliations:** 1Department of Cardiology, Arnaud de Villeneuve University Hospital, 34295 Montpellier, France; 2Department of Cardiology, Bretagne Atlantique General Hospital, 56000 Vannes, France; 3Hôpital Cardiovasculaire Louis Pradel, 69002 Lyon, France; 4Department of Cardiology, CHU Angers, Université d’Angers, 49100 Angers, France; 5Cardiology Department, CHRU de Tours, 37044 Tours, France; 6EA 4245 T2I, Université de Tours, 37044 Tours, France

**Keywords:** anti-inflammatory, atherosclerosis, drugs, myocardial infarction

## Abstract

Incidence and mortality rates for cardiovascular disease are declining, but it still remains a major cause of morbidity and mortality. Drug treatments to slow the progression of atherosclerosis focus on reducing cholesterol levels. The paradigm shift to consider atherosclerosis an inflammatory disease by itself has led to the development of new treatments. In this article, we discuss the pathophysiology of inflammation and focus attention on therapeutics targeting different inflammatory pathways of atherosclerosis and myocardial infarction. In atherosclerosis, colchicine is included in new recommendations, and eight randomized clinical trials are testing new drugs in different inflammatory pathways. After a myocardial infarction, no drug has shown a significant benefit, but we present four randomized clinical trials with new treatments targeting inflammation.

## 1. Introduction

Cardiovascular (CV) disease (CVD) incidence and mortality rates are declining, but it still remains a major cause of morbidity and mortality [1]. The most important method of preventing CVD is to encourage a healthy lifestyle, especially smoking cessation and treating hypertension. Until the beginning of the 21st century, drug treatments to slow the progression of atherosclerosis focused mainly on reducing cholesterol levels. The paradigm shift to consider atherosclerosis as an inflammatory disease led to the development of new treatments. Moreover, growing evidence that statins, a class of lipid-lowering drugs given to people with CVD, also provide pleiotropic anti-inflammatory effects creates an opportunity to test whether treating inflammation could help to prevent a cardiac event [2]. This article briefly presents the pathophysiology of inflammation and focuses attention on therapeutics targeting different inflammatory pathways of atherosclerosis and myocardial infarction (MI).

## 2. Atherosclerosis and Inflammation

### 2.1. Inflammatory Pathway in Atherosclerosis

Atherosclerosis is currently considered, at least in part, as an inflammatory condition [3]. Inflammation is considered as a risk factor for plaque rupture, generating an independent increasing risk of CV events [4]. The complex pathophysiology of inflammation in atherosclerosis has been described previously [5,6,7]. The presence of inflammatory cells in the atherosclerotic plaque, including macrophages and monocytes, is the key element. Macrophages are deeply involved in the destabilization of atherosclerotic plaques, secreting fibrous cap-degrading matrix metalloproteinase, pro-inflammatory cytokines, and pro-thrombotic tissue factor [8]. The chronic inflammation is characterized by an imbalance between pro-resolving and pro-inflammatory mediators, such as leukotrienes and damage-associated molecular pattern-mediated inflammation. Several biomarkers have been identified as prognostic markers. For example, high-sensitivity C-reactive protein (hs-CRP) is associated with higher risk of CV events in a population at high risk for CVD [9]. Other biomarkers could provide additional information but may not be available in clinical practice, such as interleukin (IL)-6, tumor necrosis factor (TNF)-α, and IL-1β [3], identifying patients with a higher risk of plaque disruption or recurrence of short-term coronary events after acute coronary syndrome (ACS). Therefore, each inflammatory pathway has been explored as a potential therapeutic target.

### 2.2. Anti-Inflammatory Biotherapies

Many pathways in inflammation have led to multiple lines of research for the development of anti-inflammatory drugs, especially through rheumatological research. Indeed, chronic inflammatory diseases, such as psoriasis or rheumatoid arthritis (RA), are deeply associated with an increase of cardiometabolic events [10]. Interestingly, CV risk dramatically rises from the start of RA [11] but returns to baseline after the ignition of Tumor Necrosis Factor (TNF) inhibitors. The CV safety of these treatments was first evaluated in patients with inflammatory disease. In a register of 983 RA patients, the age and sex-adjusted rate ratio was 0.46 (95% CI 0.25–0.85, *p* = 0.013) for anti-TNF-treated versus not treated patients [12]. In contrast, these drugs have been shown to increase clinical status (composite score improved, worsened, or unchanged on the basis of death, hospitalization for heart failure, and New York Heart Association (NYHA) class) in patients with heart failure, limiting their use in this context and more widely in the CV field [13]. Another example of a target is the chemokine (C-C motif) ligand 5 (CCL5)/C-C chemokine receptor type 5 (CCR5) axis, which plays a pivotal role in the development and progression of atherosclerotic inflammatory disease [14]. Maraviroc, a CCR5 antagonist antiretroviral drug, was evaluated in 21 human immunodeficiency virus-positive patients. The treatment significantly improved brachial flow-mediated dilatation (an endothelial function marker) by 66% (*p* = 0.002) after 24 weeks, but there were no significant changes in markers of systemic inflammation [15]. This kind of study provides mechanistic and very preliminary data, requiring larger scale confirmation.

IL-1β, the primary circulating form of IL-1, modulates smooth muscle cell proliferation, recruitment of inflammatory cells followed by leukocyte adhesion, and production of IL-6, and exerts a pro-coagulant activity [16]. At 150 mg every 3 months, canakinumab, a human monoclonal antibody targeting interleukin-1β studied in the CANTOS trial, led to a significantly lower rate of recurrent CV events (nonfatal MI, nonfatal stroke, or CV death) than placebo (8.3/100 person-years vs. 10.4/100 person-years, *p* = 0.002), independent of lipid lowering [17]. However, there was no significant difference in all-cause mortality. These studies provided important proof that inhibiting various inflammatory pathways improve the outcome of patients with established coronary atherosclerosis. Above all, CANTOS, through a large international blinded trial, confirmed the feasibility and interest in modulating the inflammatory response in patients.

### 2.3. An Old/New Anti-Inflammatory Agent: Colchicine

Colchicine is an alkaloid extracted from plants in genus Colchicum. Its utility is widely recommended for gout, familial Mediterranean fever, and pericarditis, and the drug is now largely used in these fields. Its main mechanism of action is tubulin disruption, leading to the downregulation of multiple inflammatory pathways with a reduction in neutrophil function and migration through the vascular endothelium [18]. As we found activated neutrophils in the culprit atherosclerotic plaques of patients with unstable coronary disease, their inhibition with colchicine may reduce the risk of plaque instability. Thereby, it could improve clinical outcomes in patients with stable coronary disease. For the past decade, several clinical trials have shown the potential benefit of colchicine in secondary prevention. The latest Low Dose of Colchicine 2 study, a randomized, double-blind, large-scale trial, included 5522 patients and confirmed the benefit of colchicine treatment, with the occurrence of a primary endpoint (PED) event (CV death, stroke, MI) in 187 patients (6.8%) in the colchicine group versus 264 patients (9.6%) in the placebo group [19]. The COLCOT (COLchicine Cardiovascular Outcomes Trial) was a placebo-controlled, randomized, double-blind trial that included 4745 patients within 30 days of a heart attack. With one colchicine (0.5 mg daily) and one placebo arm, the PED was a composite of CV death, resuscitated cardiac arrest, MI, stroke, or emergency hospitalization for angina leading to coronary revascularization. The results showed a significant decrease in the PED in the colchicine arm with a hazard ratio of 0.77 (95% CI 0.61–0.96) [20].

These results led the European Society of Cardiology to include colchicine in the new recommendations for secondary CV prevention [21]. For high-risks patients, a low dose of colchicine (0.5 mg once per day) may be considered (class IIb, level A).

### 2.4. Drugs under Investigation

Other therapeutic alternatives are also being studied. Some of them have failed, including a large international study with methotrexate [22]. In this trial, methotrexate did not reduce the levels of IL-1β, IL-6, or hs-CRP and was not associated with fewer CV events than placebo among patients with atherosclerosis whose condition was stable but who were at high CV risk. Importantly, this fail underlines that inhibiting inflammation by itself is not simple and could provide multiple unexpected results. Briefly, the strategy could be to target multiple pathways, such as methotrexate or colchicine, which implies a kind of black box. There are explanations and potential mainstream pathways, but the exact impacts remain to be investigated, especially in order to develop more specific approaches with fewer side effects in clinical practice. On the other hand, having only one target, such as canakinumab, could be less efficient because of biological adaptation, or even deleterious because one target could lead to many downstream effects (fatal sepsis in the case of canakinumab).

Similar to colchicine, some studies have investigated treatments targeting cytokines that are already marketed for other indications, such as hydroxychloroquine or sarilumab. Hydroxychloroquine has anti-inflammatory properties, as it decreases the activation of innate immunity by inhibiting the stimulation of Toll-like receptor [23]. In an observational study of RA patients, hydroxychloroquine was associated with a 72% decrease in the composite clinical risk of ACS, cardiac revascularization procedures, stroke, transient ischemic attack, peripheral arterial disease, and sudden cardiac death [24]. IL-6 is a pro-inflammatory cytokine involved in the stimulation of hepatic synthesis of acute phase reactants, activation of endothelial cells, and lymphocyte proliferation [25]. Studies have shown an association between high plasma IL-6 levels and the risk of CV events [26]. Tocilizumab, a human monoclonal antibody against the IL-6 receptor, was associated with a reduced risk of MI compared to abatacept (a T-cell activator modulator) in RA patients [27]. Sarilumab, another IL-6 receptor blocker, is currently being investigated to determine the effect on atheroma in RA patients (Table 1).

Others approaches are under investigation as targeting the B cells/monocytes mobilization to the injured myocardium that is associated with increased infarct size (IS) and deterioration of cardiac function in experimental model. Recently, interesting results from a phase 1/2 clinical trial, open-label trial (RITA-MI) showed the safety of a single intravenous dose of rituximab, a monoclonal anti-CD20 antibody targeted against human B cells, administered in patients with STEMI within 48 h of symptom onset. In this prospective, dose-escalation (from 200 to 1000 mg) single-arm trial, rituximab was associated with an early (within 30 min) drastic depletion of circulating B cells (mean 96.3% (93.8–98.8%)) with a maximal depletion obtained at day 6. At 6-month a dose dependent B cells repopulation without affecting immunoglobulin levels (IgG, IgM and IgA) was observed opening the door for future larger studies in the field [28].

## 3. Myocardial Infarction and Anti-Inflammatory Drugs

### 3.1. Inflammatory Storm after MI

Early and successful myocardial reperfusion is the most effective strategy for reducing the IS due to acute MI. However, restoring the blood flow leads to an important inflammatory response known as ischemia-reperfusion injuries (IRIs) [29], including no reflow lesions, myocardial sideration, and necrosis. Immediately after reperfusion, inflammatory storm is characterized by liberation of cytokines such as IL-1β and IL-6. Simultaneously, innate immunity is activated, leading to the recruitment of inflammatory cells with sequential implication of mononuclear cells, neutrophils, dendritic cells, and lymphocytes. Neutrophils are strongly involved in the initiation and resolution of inflammation, angiogenesis, and infarct recovery, whereas T lymphocytes and dendritic cells mostly contribute to ventricular remodeling [3]. Therefore, inflammation is deeply involved after acute MI to heal the injured myocardium. However, it exerts deleterious effects at the very onset of the reperfusion, worsening the infarction [30,31]. Inflammation may then contribute to increasing the cardiac remodeling that follows MI, leading to a worsening of the prognosis.

### 3.2. Biotherapies

Despite failure of a large multicenter randomized trial to prevent CV events in atheroslerotic’s patients, methotrexate was tested on post AMI with no difference based on IS evaluated by CK and troponin area under the curve. Interestingly, the authors did not observe a difference in the inflammatory markers and even observed a lower LVEF at 3-month in the methotrexate group vs. placebo (49.0% ± 14.1 vs. 56.4% ± 10.0%; *p* = 0.01) [32].

Not surprisingly, the same inflammatory pathways as atherosclerosis became therapeutic targets in MI, especially neutrophils and macrophages. Many studies have evaluated the effect of anti-inflammatory antibodies after MI with a demonstrated decrease in the inflammation but an absence of a clinical effect. The P-selectin inhibitor inclacumab was tested in a randomized trial of 544 patients with MI without ST-elevation (NSTEMI); the authors showed a non-significant trend in troponin reduction compared to placebo [33]. The VCU-ART3 study was a randomized, placebo-controlled, double-blind trial in 99 patients with ST-segment elevation MI (STEMI) who were assigned to 2 weeks treatment with anakinra (an anti-IL1 receptor antagonist) once daily (*n* = 33), twice daily (*n* = 31), or placebo (*n* = 35). The area under the curve for hs-CRP was significantly lower in patients receiving anakinra versus placebo (median, 67 [interquartile range, 39–120] versus 214 [interquartile range, 131–394] mg·day/L; *p* < 0.001), without significant differences between the anakinra arms. There were significant differences were between anakinra and placebo in the interval changes in left ventricular end-systolic volume or left ventricular ejection fraction at 12 months. However, the incidence of death or new-onset heart failure or death and hospitalization for heart failure was significantly lower with anakinra versus placebo (9.4% vs. 25.7%, *p* = 0.046 and 0% vs. 11.4%, *p* = 0.011, respectively) [34]. Recently, a pooled analysis of three early phase randomized clinical trials (VCUART (*n* = 10), VCUART2 (*n* = 30), and VCUART3 (*n* = 99)) confirmed these results in 139 patients (anakinra group: 84 patients vs. placebo: 55 patients). Treatment with anakinra significantly reduced the area under the curve for hs-CRP between baseline and 14 days (75.48 [41.7–147.47] vs. 222.82 [117.22–399.28] mg day/L, *p* < 0.001) and significantly reduced the incidence of all-cause death or new-onset HF (7 [8.2%] vs. 16 [29.1%], log-rank *p* = 0.002) and of all-cause death or HF hospitalization (0 [0] vs. 5 [9.1%], log-rank *p* = 0.007) at 1-year [34].

In a 52-week, randomized, placebo-controlled, double-blind, observational trial in patients (*n* = 151) with moderate to severe plaque psoriasis without clinical CVD, secukinumab, a human monoclonal antibody against IL-17A, was evaluated in patients after MI [35]. At week 52, bFMD was significantly higher than baseline in patients receiving the label dose of 300 mg secukinumab (+2.1%, 95% CI 0.8–3.3; *p* = 0.0022). No clinical outcome was evaluated.

The anti-IL-6 receptor antibody tocilizumab was tested in 117 NSTEMI patients versus placebo in a two-center, placebo-controlled, double-blind trial. Median AUC for hs-CRP (PED) and hs-TnT were significantly decreased during hospitalization without any safety issue [36]. The Short-Term Application of Tocilizumab Following Myocardial Infarction (STAT-MI) trial included 28 patients who received tocilizumab within the first 24 h of admission for acute MI. No significant differences in major adverse cardiac events were observed after 30 days of enrollment [37]. Recently, a larger randomized controlled trial compared 101 tocilizumab (280 mg IV) to 98 control STEMI patients included within 6 h of symptoms onset and reported a larger myocardial salvage index in the tocilizumab group (+5.6 [95%CI 0.2 to 11.3] percentage points, *p* = 0.04) and less extensive microvascular obstruction but without difference in the final IS (7.2% vs. 9.1% of myocardial volume, *p* = 0.08) [38].

Finally, elamipretide, a small mitochondrially targeted tetrapeptide (D-Arg-dimethylTyr-Lys-Phe-NH2), appears to reduce the production of toxic reactive oxygen species and stabilize mitochondrial energetics to enhance myocyte survival during IRIs in experimental studies. However, in a phase 2a multicenter randomized controlled trial (EMBRACE-STEMI), it was not associated with an improvement in IS or clinical outcomes [39].

Thus, patients may benefit from anti-inflammatory therapeutics after MI and the challenge is to find drugs which can achieve this at acceptable cost and low risk of side effects so that patients can be treated for their disease over decades.

### 3.3. From Cyclosporine to Colchicine

The final IS after an MI depends on artery occlusion time but also on reperfusion injury. Preclinical studies indicate that the opening of the mitochondrial permeability transition pore (PTP) in the inner mitochondrial membrane plays a major role in reperfusion injury [40]. Either genetic or pharmacological inhibition of cyclophilin D, a major component of the PTP, reduces the severity of IRI [41]. Cyclosporine, a pharmacological inhibitor of cyclophilin D, has been well-studied immediately after patients present with STEMI. In the Cyclosporine before PCI in Patients with Acute Myocardial Infarction (CIRCUS) trial, which included 791 patients presenting with an anterior STEMI, cyclosporine did not reduce the incidence of the separate clinical components of the primary outcome or other events, including recurrent infarction, unstable angina, and stroke, but the CIRCUS II study is ongoing for a 3-year follow-up in these patients [42]. For its potent anti-inflammatory properties via a unique mechanism of action, colchicine was also evaluated in these patients. An ancillary study of the COLCOT study (COLCOT-J3) showed the beneficial effect of early introduction, within 3 days after the infarction, as recommended in France, with a very strong event reduction effect [43]. The early effect of colchicine probably appears to be related to the reduction in post-MI inflammatory discharge, a setting in which colchicine is still being tested. The COVERT-MI trial (Effect of Colchicine on Myocardial Injury in Acute Myocardial Infarction) was a double-blind multicenter trial that assigned 192 patients to receive oral colchicine (2 mg loading dose followed by 0.5 mg twice a day) or matching placebo from admission to day 5. The PED was IS determined by cardiac magnetic resonance imaging at 5 days. There was no difference in the IS between the colchicine and placebo groups at 5 days (26 IQR [16–44] vs. 28.4 IQR [14–40] g of LV mass, respectively; *p* = 0.87) or at 3 months (17 IQR [10–28] vs. 18 IQR [10–27] g of left ventricular mass, respectively; *p* = 0.92) [44].

### 3.4. Future Anti-Inflammatory Therapies

A few drugs are currently being tested to target inflammation in MI. Matrix metalloproteinase 2 (MMP-2), a protein that cuts other proteins into pieces, is activated in injured myocardium inducing IRIs and myocardial fibrosis [45]. Blocking MMP-2 activity by doxycyclin, the only drug currently approved for clinical use, was associated with improved LV remodelling after anterior MI in a randomized open-label pilot study (TIPTOP study) with reduction in left ventricular end-diastolic (LVEDVi) and end-systolic volume index (LVESVi) (respectively, 8 mL ± 14 mL, *p* = 0.004 and −6 mL ± 12 mL, *p* = 0.02), and left ventricular ejection fraction improvement (+5% ± 12%, *p* = 0.04) [46]. A large multicenter randomized double-blind clinical trial is currently recruiting to confirm this result on LVEDVi at 90-day and explore clinical endpoints (Table 2).

Glucocorticoids, which can suppress both oxidative stresses triggering and MMP-2 activation, seems to be promising in the modulation of post-MI associated inflammation [48]. However, despite attractive acute CV physiological effects, results of corticosterioids’ use after AMI have shown conflicting results in the literature [49,50]. Some argues that individual differences in the intensity of the post-AMI inflammatory response and timing of modulation therapies used, may contribute to adverse LV remodeling and explain this conflicting results [50]. A large multicenter randomized controlled double-blind trial is ongoing to test the role of an early administration in the prehospital setting of an intravenous infusion of high dose of methylprednisolone on the IS at 3 months on CMR compared to placebo (PULSE-MI: Pulse Glucocorticoid Therapy in Patients With ST-Segment Elevation Myocardial Infarction).

IL-2 plays an important role in the control of immunity by inducing the development and survival of T regulators (Treg) cells. Among all T-cell and natural killer cell subsets, Treg cells show a much lower threshold response to IL-2 receptor signaling [51]. Therefore, in contrast to high doses, low-dose IL-2 supplementation could activate the proliferation of Treg lymphocytes while limiting the expansion of effector T cells. This hypothesis was confirmed in the LILACS trial, which tested low-dose IL-2 therapy in patients with stable ischemic heart disease and patients with ACS [52]. Treatment was associated with an increase in Treg cell count without adverse events of major concern [53]. Another phase II study (IVORY) is currently ongoing to investigate the efficacy of low-dose IL-2 injection in reducing blood vessel inflammation in patients with ACS. The mTOR pathway is also a potential therapeutic target for reducing inflammation in CVD. The CLEVER-ACS trial investigates the use of the mTOR inhibitor everolimus in patients with STEMI to determine the efficacy in reducing IS, remodeling, and clinical events in this patient population. As explained above, IL-1 represents a therapeutic target of choice to decrease inflammation after MI. RPH-104, a new TRAP molecule that inhibits both circulating IL-1ß and IL-1α, is being evaluated for its effectiveness in reducing the concentration of hs-CRP after MI.

Others prospective randomized double-blind trial are currently investigating a selective human monoclonal anti-phosphorylcholine antibody (ATH3G10) vs. placebo or the CRP apharesis vs. usual care, with primary endpoint, respectively, based on left ventricular end-diastolic change at 90-day or IS at 5-day evaluated with cardiac magnetic resonance imaging (Table 2).

### 3.5. Targeting Cell Death with Stem Cells

Following MI, cell death can occur in regulated and non-regulated forms [54].

The primary mode of myocardial cell death after MI is cell necrosis, considered with morphological criteria. On the other side, apoptosis is the first regulated cell death process. In several studies, the inhibition of apoptosis receptors and the reduction in mitochondrial apoptosis reduce apoptosis of cardiomyocytes and therefore the size of MI [55]. Necroptosis, pyroptosis, ferroptosis, parthanatos, autophagy-dependent cell death, and mitochondrial-dependent necrosis are all involved in cell death during MI [56].

Apoptosis is well-defined in two pathways, extrinsic and intrinsic, both of which are activated in cardiomyocytes under pathophysiological conditions [57]. In the exogenous pathway, cell death is induced by the activation of death domain receptors on the cell membrane, triggered by Fas ligand or tumor necrosis factor (TNF)-α. In a model of *I*/*R* heart disease, increased expression of Fas and TNF-α is associated with increased apoptosis in cardiomyocytes.

Necroptosis, a regulated mode of cell death with a necrotic appearance, can be induced by binding of TNF- α or Fas receptor ligands. Activation of the kinases Receptor-interacting serine/threonine-protein kinase (RIPK)1 and RIPK3 is the critical event in the induction of necroptosis. RIPK3 phosphorylates and then activates the necroptosis effector pseudokinase: mixed lineage kinase domain such as pseudokinase, which oligomerizes and permeabilizes the plasma membrane.

Pyroptosis is a regulated form of cell death closely tied to the innate immune response. The permeabilization of the plasma membrane and extracellular release of inflammatory cytokines is mediated by gasdermin D (GSDMD). Through undefined mechanisms, GSDMD leads to formation of pores in the plasma membrane and cause the release of intracellular material inducing cell death. The pathogen-related molecular patterns (PAMPs) or risk-related molecular patterns (DAMPs) are then detected by different inflammasomes, which are composed of nucleotidebinding oligomerization domain (NOD) and NOD-like receptors (NLR), mainly the NLRP3. In a mouse model of MI, increased NLRP3 was observed in the scar tissue and adjacent myocardial cells. Autophagy is an essential process where damaged proteins are reduced to amino-acids and fatty acids for energy generation and recycling. These metabolic processes are activated during nutrient deficiency or metabolic stress to maintain tissue and normal heart function. Two classic signaling pathways have been described: Type I PI3K-mammalian target of rapamycin (mTOR) and the pathway induced by AMP-activated protein kinase (AMPK). In a study by Sciarretta et al., autophagy showed to play a crucial role in inhibiting the occurrence and development of cardiovascular diseases such as MI, heart failure, and atherosclerosis [58]. On the other hand, Demircan et al. reported that patients with coronary heart disease or acute MI had an over-regulation of autophagy than the healthy controls [59]. Thus, over induction of the autophagy process may cause adverse effects to cells which indicates the importance of controlling the degree of autophagy in disease treatment.

In recent studies, the protective effect of the injected stem cells is not produced by direct differentiation itself but mediated by exosomes secreted from stem cells [60,61]. Thus, exosomes may be an effective modification to overcome the shortcomings of cell therapy. They are heavily involved in cell-to-cell communication by regulating various cellular processes in target cells, including adjacent cells and cells in remote parts. Several studies have reported that stem cell-derived exosomes could be used to treat ischemic disease [62,63]. The EXCELLENT (EXpanded CELL ENdocardiac Transplantation) trial is actually recruiting to evaluate the safety and the efficacy of in vitro expanded peripheral blood CD34+ stem cells injected in patients with an AMI and a LVEF remaining below 50% versus standard of care (NCT02669810).

## 4. Future in the Field of Lipid-Lowering Drugs

As well-established, statin could exert pleiotropic cardiovascular protective effects, as reviewed elsewhere [64]. The interest of lipid-lowering drugs has been reinforced recently. The EVSTEMI trial (NCT05613426) will include 330 patients to evaluate the interest of early administration of evolocumab to reduce ventricular remodeling. In a similar study with alirocumab 75 mg in 160 patients (two arms), the endpoint will be the myocardial salvage index, a usual surrogate endpoint in this kind of trials (NCT05292404). Other smaller studies will investigate the impact of early dose of alirocumab (NCT02938949) or early but also high dose of evolocumab to reduce inflammation in patients with ACS (NCT04082442). Early and high doses of anti-PCSK9 after MI seems feasible and safe in the trial EPIC-STEMI, showing a reduction in LDL-cholesterol by 72.9% with alirocumab (2.97 mmol/L to 0.75 mmol/L) vs. 48.1% with sham-control (2.87 mmol/L to 1.30 mmol/L), at a median of 45 days. Future studies are needed with clinical outcomes to evaluate more precisely the clinical implication of these approaches [65].

## 5. Conclusions

The inflammatory pathway of atherosclerosis is an emerging therapeutic target in drug development. Delaying evolution of the atherosclerotic plaque or preventing recurrence after a first CV event can have major consequences on public health.

## Figures and Tables

**Table 1 pharmaceuticals-16-00078-t001:** Ongoing clinical studies of treatments targeting inflammation in the context of atherosclerosis.

Trial Name	Study Design	Study Population	Estimated Enrollment Estimated Completion Date	Intervention	Target	Primary Outcome	Clinical Trials Identifier
CHANGAN	Phase IV, single center, double-blind, randomized, placebo-controlled	Patients with CAD and hs-CRP >1 mg/L	35 participants Completed	Hydroxychloroquine	Broad immunosuppression	Change in fasting hs-CRP level	NCT02874287
	Phase IV, multicenter, blind, placebo-controlled	Asymptomatic for atherosclerotic disease aged 30 to 60 years and exposed to air pollution	200 participants 2023	Montelukast	Leukotriene receptor antagonist	Subclinical atherosclerosis defined as changes in brachial flow-mediated dilation and carotid intima media thickness	NCT04762472
SARIPET	Phase IV, single-center, open label	Patients with active rheumatoid naïve to biological DMARDs or refractory to a single biological other than anti-IL-6 drugs	20 participants Unknown	Sarilumab	IL-6 receptor blocking monoclonal antibody	Change in carotid atheroma plaque assessed by ultrasonography	NCT04350216
PAC-MAN	Phase II, randomized, double-blind, placebo- controlled	Patients with stable CAD	40 participants Unknown	Paclitaxel	Blocks cellular proliferation (antimicrotubule agents)	Reduction in plaque size measured by coronary CTA from baseline to 6–8 months	NCT04148833
ZEUS	Phase III, multicenter, double-blind, randomized, placebo-controlled	Patients with CKD stage 3 to 4, known ASCVD, and hs-CRP >2 mg/L	6200 participants 2025	Ziltivekimab	IL-6 blocking monoclonal antibody	Time to first occurrence of MACE	NCT05021835
Lp(a) HORIZON	Phase III, multicenter, double-blind, randomized, placebo-controlled	Patients with established CVD and Lp(a) ≥70 mg/dL	7680 participants 2025	Pelacarsen	Antisense oligonucleotide targeting Apo(a)	Time to first occurrence of expanded MACE in patients with Lp(a) ≥ 70 mg/dL or Lp(a) ≥ 90 mg/dL	NCT04023552
	Phase II/III, multicenter, double-blind, randomized, placebo-controlled	Patients with multivessel CAD and hs-CRP >2 mg/L	40 participants 2023	Methotrexate delivered in LDL-like nanoparticles	Dihydrofolate reductase inhibitor	Change in plaque volume measured by CTA	NCT04616872
GOLDILOX	Phase IIB, multicenter, double-blind, randomized, placebo-controlled	Patients aged ≥21 years with a history of MI and hs-CRP >1 mg/L	400 participants 2023	MEDI6570	LOX-1 receptor blocking monoclonal antibody	Change in non-calcified plaque volume measured by CTA	NCT04610892

ACS: acute coronary syndrome; ASCVD: atherosclerotic cardiovascular disease; CAD: coronary artery disease; CKD chronic kidney disease; CTA: computed tomography angiography; DMARD: disease modifying anti-rheumatic drug; IL: interleukin; Lp(a): lipoprotein (a); MACE: major adverse cardiovascular event.

**Table 2 pharmaceuticals-16-00078-t002:** Ongoing clinical studies of treatments targeting inflammation in post-myocardial infarction.

Trial Name	Study Design	Study Population	Estimated Enrollment Estimation Completion Date	Intervention	Target	Primary Outcome	Clinical Trials Identifier
PULSE-MI	Randomized, multicenter, double-blind, placebo-controlled clinical trial	Patients with STEMI	400 participants 2027	Methylprednisolone 250 mg IV in prehospital setting	Ischemia-reperfusion injury prevention and wide anti-inflammatory effect	Infarct size measured by late-gadolinium enhancement on CMR at 90-day	NCT05462730
IVORY	Phase II, randomized, double-blind, placebo-controlled, parallel group	Patients with ACS or UA who have hsCRP >2 mg/L	60 participants 2024	Low dose IL-2	Induces expansion of regulatory T cells	Change in vascular inflammation measured by mean TBRmax in the index 18F-FDG PET/CT	NCT04241601
CLEVER-ACS trial	Phase II multicenter, double-blind, randomized, placebo-controlled	Patients with ACS undergoing PCI (randomization within 5 days after PCI)	150 participants Completed	Everolimus (7.5 mg for 3 days, followed by 5 mg for 2 days)	Selective mTOR inhibitor	Change in myocardial infarct size from baseline as measured by MRI at 30-day follow-up	NCT01529554
_	Phase II, multicenter, double-blind, randomized, placebo-controlled	Patients with STEMI undergoing PCI	102 participants 2022	RPH-104	IL-1α/IL-1β inhibitor heterodimeric fusion protein	hsCRP AUC from baseline until day 4	NCT04463251
anaRITA MI2	Phase II multicenter, double-blind, randomized, placebo-controlled	Patients with STEMI	558 participants Unknown	Rituximab	B-cell depletion with CD20	LVEF at 6 months with cardiac magnetic resonnance	NCT05211401
_	A Phase IIa, Placebo-controlled, Double Blind, Randomized Multicenter Pilot Study	Patients with STEMI	82 patients Completed	Human monoclonal antibody (ATH3G10)	Decrease in phosphorylcholine mediated inflammation	Change in LVEDVi at 90-day	NCT03991143
_	A randomized, open-label, controlled, multicenter, two group trial	Patients with STEMI	202 patients 2022	PentraSorb^®^-CRP apharesis performed at day 1, 2 and 3 post PCI	CRP apheresis	Infarct size visualized by CMR at 5 ± 2 days post PCI [47]	NCT04939805
_	Randomized, double-blinded, placebo-controlled study	Patients with STEMI	170 participants 2023	Doxycyclin (200 mg po and then 100 mg × 2/day during 7 days)	MMP-2 blockage	LVESVi measured by CMR at 90-day	NCT03508232

ACS: acute coronary syndrome; AUC: area under the curve; CMR, cardiac magnetic resonance; hs-CRP: high-sensitivity C-reactive protein; IL-1: interleukin-1; IL-2: interleukin-2; LVEDi, Left Ventricular End-Diastolic Volume index; LVESVi, Left ventricular end-systolic volume index; MMP-2, Matrix metalloproteinase 2, PCI: percutaneous coronary intervention; STEMI: ST-segment elevation myocardial infarction; TBR: target-to-blood pool ratio, UA: unstable angina.

## Data Availability

Not applicable.

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
