# Peer review of "Anti-Inflammatory Drug Candidates for Prevention and Treatment of Cardiovascular Diseases"

_pharmaceuticals, 2023, doi:10.3390/ph16010078_

Round 1

Reviewer 1 Report

The manuscript “Anti-inflammatory Drug Candidates for Prevention and Treatment of Cardiovascular Diseases” by Delbaere Q et al., discuss the pathophysiology of inflammation with a particular attention on therapeutics targeting different inflammatory pathways of atherosclerosis and myocardial infarction. This manuscript is well designed and organized. The focus of this manuscript is new and original. My suggestion would be to add a final paragraph discussing the potential future therapies in preclinical development (myeloid cells, NETs, trained immunity, inflammosome, Efferocytosis, etc..).

Author Response

We would like to thank the reviewer for his positive evaluation of our work.

We agree this suggestion would be of interest and we have added a new paragraph from page 9 to 10:

“ 3.5. Targeting cell death with stem-cells

Following MI, cardiac I/R injury, heart failure, and other heart diseases, cell death can occur in regulated and non-regulated forms [54].

Morphological criteria consider that cell necrosis is the primary mode of myocar-dial cell death after MI. In contrast, apoptosis is considered to be the first regulated cell death process. Several studies have shown that the inhibition of apoptosis receptors and the reduction of mitochondrial apoptosis reduce apoptosis of cardiomyocytes and reduce the size of MI [55]. Necroptosis, pyroptosis, ferroptosis, parthanatos, autopha-gy-dependent cell death, and mitochondrial-dependent necrosis are all involved in cell death during MI [56].

Apoptosis is mediated by two well-defined pathways, extrinsic and intrinsic, both of which are activated in cardiomyocytes under pathophysiological conditions [57]. In the exogenous pathway, cell death is induced by the activation of death domain recep-tors on the cell membrane. It is triggered by Fas ligand or tumor necrosis factor (TNF)-α. Increased expression of Fas and TNF-α is associated with increased apoptosis in cardiomyocytes in a model of I/R heart disease.

Necroptosis, a regulated mode of cell death with a necrotic appearance, can be in-duced by binding of TNF- α or Fas receptor ligands. The critical event in the induction of necroptosis is the activation of the kinases Receptor-interacting serine/threonine-protein kinase (RIPK)1 and RIPK3. RIPK3 phosphorylates and then activates the necroptosis effector pseudokinase : mixed lineage kinase domain like pseudokinase which oligomerizes and permeabilizes the plasma membrane.

Pyroptosis is a regulated form of cell death that is closely tied to the innate im-mune response. The permeabilization of the plasma membrane and extracellular re-lease of inflammatory cytokines is mediated by gasdermin D (GSDMD). Through still undefined mechanisms, GSDMD leads to formation of pores in the plasma membrane and cause the release of intracellular material inducing cell death. The pathogen-related molecular patterns (PAMPs) or risk-related molecular patterns (DAMPs) are then detected by different inflammasomes, which are composed of nucleotidebinding oligomerization domain (NOD) and NOD-like receptors (NLR), mainly the NLRP3. In a mouse model of AMI, increased NLRP3 was observed in scar tissue and the cyto-plasm of adjacent myocardial cells.  Autophagy is an essential process where damaged proteins are broken down into amino-acids and fatty acids for energy generation and recycling. These metabolic processes are activated during nutrient deficiency or meta-bolic stress to maintain tissue function, a process essential for maintaining normal heart function. Two classic signaling pathways from the entire autophagy signaling network have been described : Type I PI3K-mammalian target of rapamycin (mTOR) and the pathway induced by AMP-activated protein kinase (AMPK). A study by Sciar-retta et al. showed that autophagy plays a crucial role in inhibiting the occurrence and development of cardiovascular diseases such as MI, heart failure, and atherosclerosis [58]. On the other hand, overinduction of the autophagy process may cause adverse ef-fects to cells which indicates the importance of controlling the degree of autophagy in-duction in disease treatment. Thus, autophagy seems to be a double-edged sword in the treatment of MI. Demircan et al. also reported that patients with coronary heart dis-ease or acute MI had an over-regulation of autophagy than the healthy controls [59].

In recent studies, the myocardial protective effect of the injected stem cells is not produced by direct differentiation itself but mediated by exosomes secreted from stem cells [60,61]. Thus, exosomes may be an effective alteration to overcome the shortcom-ings of cell therapy. They play an essential role in cell-to-cell communication by regu-lating various cellular processes in target cells, including adjacent cells and cells in re-mote parts. Several studies have reported that stem cell-derived exosomes could be used to treat ischemic disease [62,63]. The EXCELLENT (EXpanded CELL ENdocardiac Transplantation) trial is actually recruiting to evaluate the safety and the efficacy of in vitro expanded peripheral blood CD34+ stem cells injected in patients with an AMI and a LVEF remaining below 50% versus standard of care (NCT02669810).”

Reviewer 2 Report

The authors address a topic of great practical interest, however, in literature there are already similar reviews. For this reason, I think that although the manuscript is well written, it is poor in novelty information to be considered for publication.  

Adding details on the molecular mechanisms underlying cardiovascular prevention could attract the readers' attraction more.

Author Response

We would like to thank the reviewer for his positive evaluation of our writing. Although there are already reviews in the field, we have tried to propose an original view and some propositions to go a step further.

We have tried to develop some molecular aspects and promising new avenues in the last paragraph, but it is difficult to reach the good deal between futile details and reliable background.

Reviewer 3 Report

Cardiovascular disease remains a major cause of morbidity and mortality. A lot of data suggest that atherosclerosis as an inflammatory disease. The authors had the excellent idea to elaborate a review on the medication with anti-inflammatory effect on the atheroma plaque. This article briefly presents the pathophysiology of inflammation and focuses attention on therapeutics targeting different inflammatory pathways of atherosclerosis and myocardial infarction.

The article is very well written. I would still have a proposal for authors:  the development of new treatments (like monoclonal antibody to proprotein convertase subtilisin/kexin type 9) and the growing evidence that statins also provide pleiotropic anti-inflammatory effects can be also discussed in this manuscript? How do you interprets the results of EPIC STEMI trial? 

Author Response

We would like to thank the reviewer for his very positive evaluation of our work.

A new final paragraph has been included to present new avenues in the field of lipid-lowering drugs, including the proprotein convertase subtilisin/kexin type 9. This paves the way for future studies on the interest to treat very soon patients with treatments able to lower deeply cholesterol. From page 10 to 11.

“4. Future in the field of lipid-lowering drugs

As well-established, statin could exert pleiotropic cardiovascular protective ef-fects, as reviewed elsewhere [64]. The interest of lipids-lowering drugs has been rein-forced recently. The EVSTEMI trial (NCT05613426) will include 330 patients to evalu-ate the interest of early administration of evolocumab to reduce ventricular remodel-ing. In a similar study with alirocumab 75 mg in 160 patients (two arms), the endpoint will be the myocardial salvage index, a usual surrogate endpoint in this kind of trials (NCT05292404). Other smaller studies will investigate the impact of early dose of ali-rocumab (NCT02938949) or early but also high dose of evolocumab to reduce inflam-mation in patients with ACS (NCT04082442). Early and high doses of anti-PCSK9 after MI seems feasible and safe in the trial EPIC-STEMI, showing a reduction of LDL-cholesterol by 72.9% with alirocumab (2.97 mmol/L to 0.75 mmol/L) vs. 48.1% with sham-control (2.87 mmol/L to 1.30 mmol/L), at a median of 45 days. Future studies are needed with clinical outcomes to evaluate more precisely the clinical implication of these approaches [65].”

Round 2

Reviewer 2 Report

I really appreciate the new paragraph, adding novelty to the manuscript. I think that now can be published without other revision.

Author Response

We are pleased that this new paragraph is suitable for you.